



# SHEDIS-Temperature: Linking temperature-related disaster impacts to subnational data on meteorology and human exposure

Sara Lindersson [1,2,3], Gabriele Messori [1,3]

1. Department of Earth Sciences, Uppsala University, Uppsala, 75236, Sweden
2. Centre of Natural Hazards and Disaster Science (CNDS), Sweden
3. Swedish Centre for Impacts of Climate Extremes (climes), Sweden

*Correspondence to*: Sara Lindersson (sara.lindersson@geo.uu.se)

**Abstract**

International databases of disaster impacts are crucial for advancing disaster risk research, particularly as climate change
intensifies the frequency and intensity of many natural hazards – including temperature extremes. However, many widely-used disaster impact databases lack information on the physical dimension of the hazards associated with an impact, and on the exposure to such hazards. This hinders analysing drivers of severe disaster outcomes. To bridge this knowledge gap, we present SHEDIS-Temperature, a dataset that provides Subnational Hazard and Exposure information for temperature-related DISaster impact records (https://doi.org/10.7910/DVN/WNOTTC ; Lindersson and Messori, 2025). This open-access dataset
links temperature-related impact records from the Emergency Events Database (EM-DAT) with subnational data on their locations, associated meteorological time series, and population maps. SHEDIS-Temperature provides hazard and exposure data for 2,835 subnational locations associated with 382 disaster records from 1979 to 2018 in 71 countries. Detailed hazard metrics, derived from 0.1° 3-hourly data, encompass absolute indicators, such as the heat stress measure apparent temperature accounting for humidity and wind speed, as well as percentile-based indicators of when and where temperatures exceeded
local thresholds. Population exposure data include annual population figures for impacted subnational administrative units and person-days of exposure to threshold-exceeding temperatures. Outputs are available at grid-point level as well as zonally aggregated to administrative subdivision units, and disaster-record levels. By providing comprehensive attributes across the hazard-exposure spectrum, SHEDIS-Temperature supports interdisciplinary research on past temperature-related disasters, offering valuable insights for future risk mitigation and resilience strategies.

## 1 Introduction

International databases documenting impacts from natural hazards play a central role in advancing quantitative research on disaster risk (Jones et al., 2022; UNDRR, 2022). These collections enable researchers, organisations and agencies to track how disaster impacts vary across regions and over time, including to monitor progress of disaster risk reduction globally (Aitsi-Selmi et al., 2015). Combined with additional risk-relevant information – such as estimates of the physical hazard, exposed





population, and socioeconomic indicators – impact records can pinpoint factors contributing to particularly severe outcomes (Kahn, 2005; Lindersson et al., 2023; Mochizuki et al., 2014; Tselios and Tompkins, 2019; Vestby et al., 2024). Lessons learned from past events are, furthermore, increasingly important for guiding future risk mitigation and resilience efforts as climate change drives shifts in the frequency and intensity of many natural hazards (IPCC et al., 1990). Hot and cold extremes are primary examples of fatal hazards under rapid change (Gallo et al., 2024; García-León et al., 2024; Gasparrini et al., 2015;
IPCC, 2023; Lüthi et al., 2023; Russo et al., 2019).

The Emergency Events Database (EM-DAT - CRED and UCLouvain, 2024a), maintained by the Centre for Research on the Epidemiology of Disasters, is a leading open-access resource for international disaster impact data (Jones et al., 2022; Panwar and Sen, 2020). Widely used for its extensive set of national-level records of human and economic losses from major disasters, EM-DAT remains a cornerstone of empirical disaster research despite certain limitations, such as underreporting and a bias
toward advanced economies (Acevedo, 2016; Green et al., 2019; Jones et al., 2022, 2023; Wirtz et al., 2014). Beyond issues with missing data, EM-DAT lacks spatiotemporal detail for impacts and the associated hazards. The physical magnitude of temperature extremes, for instance, is often missing or reduced to a single maximum or minimum air temperature, without specifics on the timing, duration or location. Furthermore, multiple meteorological factors beyond the (dry bulb) air temperature, including humidity and wind, substantially influence stress levels experienced by the human body during extreme
temperatures (Cvijanovic et al., 2023). These limitations place the responsibility on users to link impact records to additional data sources when a more comprehensive risk analysis is needed. However, recent advancements in high-resolution data on disaster locations, meteorological data and population patterns present new opportunities for systematic data integration across the risk spectrum.

This article introduces SHEDIS-Temperature, an open-access dataset that provides S̲ubnational H̲azard and E̲xposure
information for temperature-related DIS̲aster impact records (Figure 1). To achieve this, we integrated the open-source Geocoded Disasters extension (GDIS; Rosvold and Buhaug, 2021), which geocodes many EM-DAT records to subnational locations, with high-resolution global time-series of meteorological variables from Multi-Source Weather (MSWX; Beck et al., 2022) and population data from the Global Human Settlement Population grids (GHS-POP; European Commission, 2023; Schiavina et al., 2023). SHEDIS-Temperature (Lindersson and Messori, 2025) provides hazard and exposure data for 2,835
subnational locations (referred to hereafter as *subdivisions*) associated with 382 disaster records from 1979 to 2018 in 71 countries (Figure 1).

SHEDIS-Temperature advances a growing field of research that links disaster impact data to in-situ and satellite-derived information (Brimicombe et al., 2021; Dellmuth et al., 2021; Felbermayr and Gröschl, 2014; Kageyama and Sawada, 2022; Mester et al., 2023). Our dataset offers three primary contributions. First, it includes detailed information on the physical
hazards, including both absolute and percentile-based indicators. The absolute indicators, like maximum 2-m air temperature and apparent temperature are provided as daily statistics derived from 3-hourly data. The percentile-based threshold analysis



identifies if, when, where, and by how much daily temperatures exceeded local 90th and 95th percentiles, enabling more context-sensitive assessments of extreme events. Second, SHEDIS-Temperature provides data on population exposure to these extreme temperatures, detailing annual population figures for each impacted subdivision and exposure to threshold-exceeding

temperatures, expressed as person-days. Third, to support diverse research needs, we present outputs at three levels: grid point, subdivision and EM-DAT record (referred to hereafter as *disno*, short for disaster number). Open-access source scripts enable users to further adjust the outputs if needed.

The usefulness of SHEDIS-Temperature is multifaceted. It can serve as a corroboration of EM-DAT and GDIS by cross-verifying reported impact locations against observed extreme weather events. We also anticipate that SHEDIS-Temperature

can support empirical analysis of temperature-related disasters across disciplines. We consider the granularity and flexibility of the dataset to be crucial, especially since disasters often have uneven impacts – not only across countries but within them as well (Masselot et al., 2023; Yin et al., 2023). Our work is also aligned with UNDRR's call for more integrated tracking systems that capture both the origins of hazards and their impacts (UNDRR, 2022). Ultimately, systematically connecting data on hazards, exposure and impacts is essential for quantifying the social vulnerability to disasters.

**Figure 1 Introduction to SHEDIS-Temperature. (a)** The 382 national-level disaster records from 1979 to 2018 in 71 countries that underpin the dataset. These records comprise EM-DAT records **(b)** that have been geocoded to administrative subdivisions by GDIS **(c)**. SHEDIS-Temperature expands on this by providing an extensive catalogue of hazard- and exposure-related attributes for each subdivision and disno **(d)**, as well as data on grid point level.

## 2 Data and methods

SHEDIS-Temperature links temperature-related impact records to subnational data on their physical occurrence and human exposure. The dataset was constructed through three main steps: (1) sampling and geocoding, (2) data processing at the grid point level, and (3) aggregating of outputs into the final dataset (Figure 2). All analyses were performed in R v.4.3.3 and the WGS84 coordinate system. Meteorological data were processed with Climate Data Operators (Schulzweida, 2023).




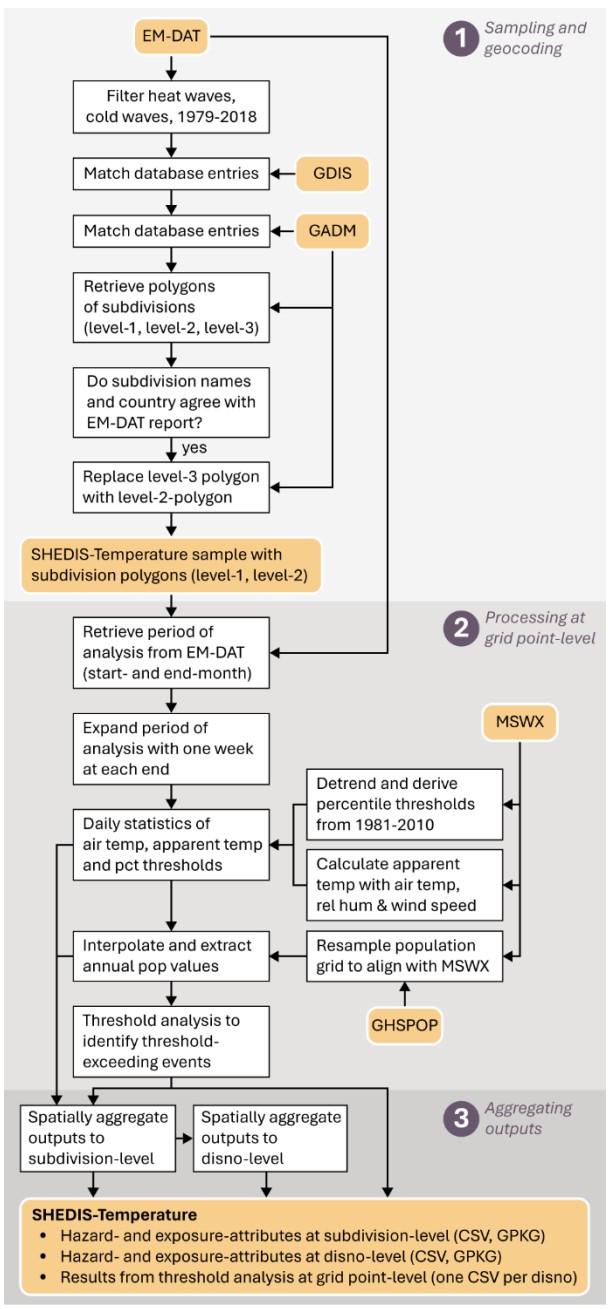


**Figure 2. Flowchart illustrating the main steps of integrating data from multiple sources to derive SHEDIS-Temperature.** (**1**) A total of 382 temperature-related impact records from EM-DAT were successfully matched to subnational locations by GDIS, which we used to identify 2,835 subdivision boundaries at level 1 (province/equivalent) and level 2 (county/district/equivalent) from GADM. (**2**) Within these identified geographical extents, daily statistics of meteorological variables (absolute values and percentiles) were computed from MSWX. Annual population figures were also interpolated from GHS-POP, and percentile-exceeding temperature events were identified. (**3**) Outputs were exported at three levels: grid point, subdivision and disno.




## 2.1 Sampling and geocoding

SHEDIS-Temperature extends the international disaster database EM-DAT (CRED and UCLouvain, 2024a), which documents national-level disaster impacts meeting at least one of the following criteria: ≥10 fatalities, ≥100 affected individuals, a declared state of emergency, and/or a request for international assistance. SHEDIS-Temperature includes records that EM-DAT classifies as heat waves and cold waves. EM-DAT defines a heat wave as a period of abnormally hot and/or unusually humid weather, while a cold wave is a period of abnormally cold weather that may be exacerbated by high winds (CRED and UCLouvain, 2024b). Both types of events are also described as typically lasting two or more days, and specific temperature thresholds vary by region (CRED and UCLouvain, 2024b).

Our dataset incorporates records from 1979 to 2018, aligning with the temporal scope of supporting datasets – Multi-Source Weather (MSWX - Beck et al., 2022) and the Global Human Settlement Population grid (GHS-POP - European Commission, 2023; Schiavina et al., 2023), which begin in 1979 and 1975 respectively, and the Geocoded Disasters (GDIS) dataset (Rosvold and Buhaug, 2021), reaching up to 2018. Limiting the dataset to four recent decades also enhances data reliability, since impact records from earlier periods are generally more uncertain and biased (CRED and UCLouvain, 2024b; Gall et al., 2009). The final sample of SHEDIS-Temperature includes impact records that meet this time-span and have been geocoded to administrative subdivisions by GDIS.

The creators of GDIS geocoded EM-DAT entries (disnos) to subnational levels by matching their location description to administrative subdivision names in GADM version 3.6 (www.gadm.org), a global database of administrative boundaries. GADM provides subdivisions across multiple hierarchical levels, including level-1 (province/equivalent), level-2 (county/district/equivalent), and level-3 (municipality/equivalent). GDIS linked each disno to one or more of these subdivisions, across one or more levels, when the location description in EM-DAT was sufficient to do so.

For each disno, GDIS provides the original location description from EM-DAT along with the name, level and centroid for one or more matched subdivisions. We identified several mismatches where a disno has been linked to a subdivision in the wrong country due to shared subdivision names. To address this, we derived country-specific ISO codes directly from the GDIS-provided coordinates and used them to reconstruct the disno number. If the assigned location falls outside the expected country, the disno number does not match with the list from EM-DAT and is excluded from the final sample. We further verified the accuracy of the centroids of the matched subdivisions.

To retrieve the boundary polygons of the GDIS subdivisions, we first converted GADM polygons to centroids and then applied a nearest-neighbour approach for each administrative level separately. We then controlled for discrepancies between the original location description in EM-DAT and the matched subdivision name. We identified one mismatch where the centroid





of a subdivision was misplaced, which we corrected manually[1]. After having run these consistency checks, we replaced the centroids from GDIS with the original subdivision boundary polygons using a GADM-specific identifier.

For the purpose of SHEDIS-Temperature, we chose to replace the level-3 impacted subdivisions (N=148, associated with 45 unique disnos) with their parent level-2 divisions – due to the relatively coarse resolution of the global supporting datasets and
the wide spatial extent of temperature extremes. Moreover, GADM provides level-3 subdivisions for only certain countries, making level-1 and level-2 divisions more suitable for consistent cross-country comparisons. Duplicates, due to multiple level-3 units having been replaced with the same level-2 unit for the same disno, were removed. To reduce file size, we simplified the polygon shapes with the R package *rmapshaper* (Teucher et al., 2023), which performs topologically aware polygon simplifications.

**2.1.1 Sample of disaster records and their subnational locations**

The final dataset comprises 2,835 impacted subdivisions, including 2,353 level-1 administrative units (province/equivalent) and 482 level-2 units (county/district/equivalent), linked to 382 distinct disaster records (disnos) across 71 countries (Figure 1). Of these, 63% of the subdivisions are linked to 243 cold wave records in 60 countries, while the remaining subdivisions are linked to 139 heat wave disnos in 47 countries. The majority (83%) of impacted subdivisions are level-1 administrative
units. Since several subdivisions experienced multiple events during the study period, the dataset includes 931 unique level-1 and 343 unique level-2 subdivisions.

The dataset spans 1979–2018, with most disnos recorded after 2000 (Figure 3a). A notable spike in cold wave records appears in 2012, when cold waves in Europe led to recorded disasters in 26 countries, ten of which recorded events both at the beginning and end of the year. The European heat waves of 2003 and 2007 are also evident (Figure 3a), resulting in 15 and 11 disaster
impact records, respectively. Each disno in the sample includes a reported start month from EM-DAT, collectively illustrating the seasonal variability of these hot and cold extremes across continents (Figure 3b).

The geographic distribution of the final sample reflects a bias in the parent dataset EM-DAT, with most records originating from Europe, Asia and the Americas (Figure 1a, Figure 3b). This bias arises from two factors: a reporting tendency towards advanced economies and the high density of small countries in continental Europe, which leads to multiple national-level
records per meteorological event.

The median number of subdivisions impacted by each disno is four for cold waves and three for heat waves, though this also varies across continents and recording periods (Figure 3c), from six in Europe to four in the Americas, three in Asia and Africa and 1.5 in Oceania. More recent records tend to be linked to a greater number of subdivisions compared to older ones, likely

---

[1] For *disno* 1999-0068-RUS, the centroid of the Russian subdivision Chukot had been incorrectly located within the neighbouring subdivision Sakha by GDIS. This error may arose because Chukot spans the 180th meridian, which can distort the centroid location depending on the methodology used.

reflecting increased detail in disaster reporting over time. Figure 3c also displays an outlier in the sample with 78 linked

subdivisions – a heat wave disno from Turkey (disno 2000-0381-TUR) for which EM-DAT offers an unusually long list of

impacted locations.

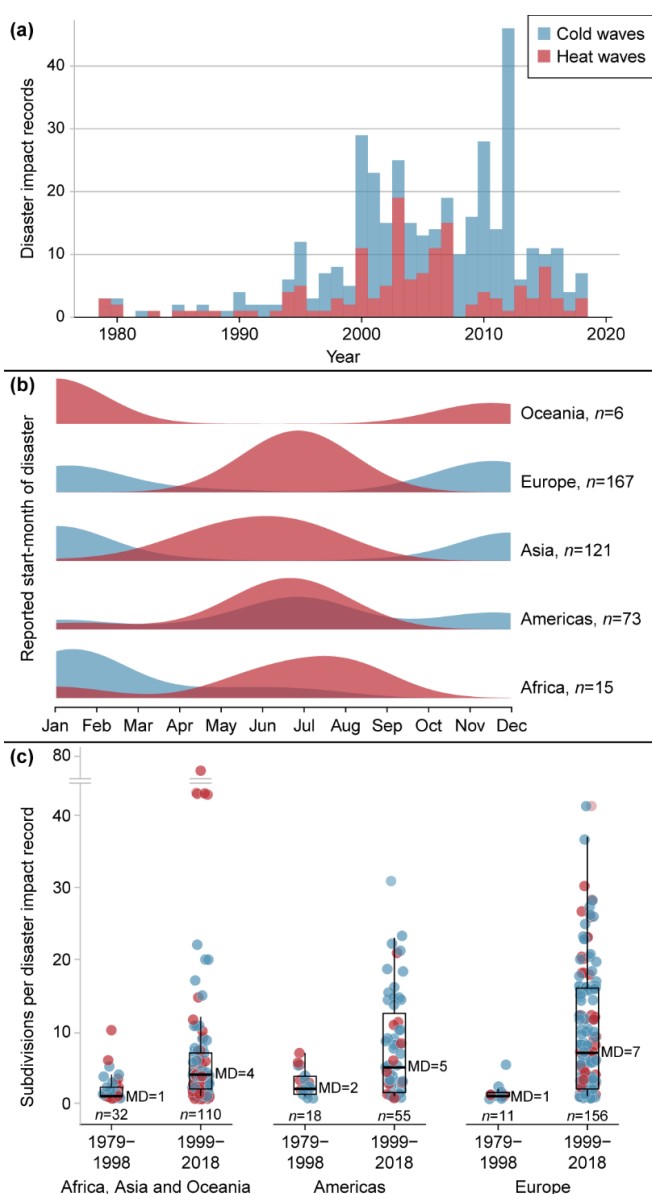

**Figure 3. Overview of the disaster records underpinning SHEDIS-Temperature. (a)** Histogram of the temporal distribution of disaster
impact records from 1979 to 2018, highlighting the higher number of records in the latter part of the recording period. **(b)** Density plots
illustrating the seasonal variability of reported start months for heat wave and cold wave records across continents. The uneven number of
observations per continent also highlight the geographic bias of the parent dataset, EM-DAT. **(c)** Boxplots showing the number of
subdivisions linked to each disno, per continent and reporting period. Note the cut in the y-axis for visualization purposes.



## 2.2 Data processing at grid point level

### 2.2.1 Spatiotemporal boundaries for analysis

The simplified polygons outlining the impacted subdivisions define the spatial boundaries for the subsequent analysis, which we also refer to as the *geometry* for analysis. The analysis period for each disno is defined by the first date of the start month and last date of the end month as reported by EM-DAT, but expanded with one week at both ends for precaution. A majority (*n*=175) of the disnos have reported start months that coincide with the reported end month, resulting in a roughly six weeks

analysis period. One impact record (disno 2007-0673-ROU) missed a reported end month, which we then assumed to be the month following the reported start month.

We chose not to rely on the daily information from EM-DAT because only approximately half of the disnos provide start and end days. Additionally, about a quarter of the records with daily information have start- and end-days that coincide, which contradicts the very disaster definition stating that heat waves and cold waves typically last for two days or more (CRED and

UCLouvain, 2024b).

### 2.2.2 Meteorological data processing

Multi-Source Weather (MSWX; Beck et al., 2022) is a high-resolution meteorological dataset derived from hourly ERA5 reanalysis data (Hersbach et al., 2020). MSWX bias-corrects and downscales the ERA5 data using nearest-neighbour interpolation to a spatial resolution of 0.1°. It provides seamless global NetCDF files at 3-hourly intervals beginning January

1, 1979. The 3-hourly MSWX values represent averages of the 1-hourly ERA5 data (Beck et al., 2022). ERA5 is widely regarded as the most reliable reanalysis dataset available. For instance, Liu et al. (2024) recently demonstrated its consistent quality for 2-m air temperature across most regions across the globe. For this study, we used MSWX-Past data on 2-m air temperature (°C), 2-m relative humidity (%) and 10-m wind speed (m/s).

Using 3-hourly values of air temperature, wind speed, and relative humidity from MSWX, we calculated apparent temperature

for each grid point within the impacted subdivisions and their respective analysis period. Apparent temperature quantifies the amplification of percieved temperatures due to wind and humidity, and can thus be used as a metric for thermal stress in humans (Steadman, 1984). The model assumes that the temperature is experienced outdoors but not in direct sunlight (Buzan et al., 2015). Although radiation is sometimes included in these calculations, we used the non-radiant version for our analysis, following the methodology by Steadman (1994). These calculations are performed using the *apparentTemp* function in the R

package *HeatStress* (Casanueva, 2019).

For each disno and subdivision in the sample, we then compiled daily time series for all grid points within the boundary polygon and the analysis period. The following variables were included: daily mean air temperature (t), daily maximum air temperature (tx), daily minimum air temperature (tn), daily mean apparent temperature (at), daily maximum apparent



temperature (atx), and daily minimum apparent temperature (atn). The variables tx, tn, atx, and atn thus represent the most

extreme 3-hourly values recorded within each 24-hour period.

### 2.2.3 Population data processing

For estimating human exposure to extreme temperatures, we used global population maps from the Global Human Settlement Layer R2023A (GHS-POP; European Commission, 2023; Schiavina et al., 2023), which combines satellite imagery and census data to generate 5-year time series from 1975 to 2020. The population maps, initially at a spatial resolution of 30 arcseconds,

were resampled to align with the MSWX 0.1° data grid using the R package *exactextractr* (Baston, 2023).

During resampling, we extracted population estimates for each 5-year time step for all grid points within the boundaries of each subdivision. These population values were simultaneously scaled with the coverage fraction, which represents the proportion of each grid point being located within the subdivision boundaries. To address minor discrepancies in population values introduced during resampling, we also scaled the resampled population cell values to ensure that the population sum

across each subdivision polygon matched the original, non-resampled population sum. Finally, we generated annual population estimates for each grid point by linearly interpolating the 5-year population estimates.

### 2.2.4 Percentiles of air temperature

We used air temperature to calculate percentiles for each day of the year relative to the 30-year reference period 1981–2010. This reference period was selected since it best corresponds with our period of analysis (1979-2018), ensuring consistency

with the temporal coverage of the study. For heat waves, we derived the 90th and 95th percentiles of daily maximum temperature. For cold waves, we calculated the 10th and 5th percentiles of daily minimum temperature. We calculated percentiles centred on a 31-day moving window (following e.g. Russo et al. (2014, 2017)), and thus we extended the reference period to also include the last 15 days of 1980 and the first 15 days of 2011 prior to the percentile calculations. February 29 was assigned the percentile value of February 28 in leap years.

We also linearly detrended the time series of daily maximum and minimum temperatures before calculating the percentiles, to remove the potential influence of long-term trends. This is in line with the notion that a climatological period should ideally be uniform (WMO Climatological Normals, 2025). We did this by using the Climate Data Operators (CDO) function *detrend* (Schulzweida, 2023). To preserve baseline characteristics, the temporal mean of the original daily maximum and minimum time series was added back to the detrended values.

### 2.2.5 Event detection analysis

We identified heat wave and cold wave events using percentile-based threshold analysis at grid point-level (Figure 4). Heat waves were detected when the daily maximum air temperature exceeded the 90th or 95th percentile thresholds (referred to hereafter as pct90 and pct95 events), while cold waves were identified when the daily minimum air temperature fell below the

10th or 5th percentiles (referred to hereafter as pct10 and pct05 events). The identification of percentile exceedances was performed on the original (non-detrended) time series to maintain consistency with the rest of the analysis, which also relied on the non-detrended meteorological data. Moreover, the percentiles are relative to the selected reference period. Consequently, the final number of identified percentile exceedances might deviate from the specified percentile numbers.

We consider the start of an event to be the first day the temperature crossed the threshold, and its end to be the first day it no longer did. If a non-qualifying day was directly preceded and followed by threshold-surpassing days, it is treated as also being threshold-surpassing, as exemplified in Figure 4.

We used a *minimum duration* of three days for a sequence to be classified as a heat wave or cold wave. We define the event *duration* as the number of days between its start and end. Event *magnitude* was calculated as the sum of temperature exceedances relative to the threshold over the event duration, following e.g. Brown (2020). Human exposure, expressed in *person-days*, was quantified by multiplying the population count at each grid point by the event duration. For example, if a grid point with a population count of 1,000 people experienced a seven-day event (as illustrated in Figure 4), the total event exposure would be 7,000 person-days.

Event-specific metrics were stored in CSV files: with one file per disno and one row per event at the grid point level.

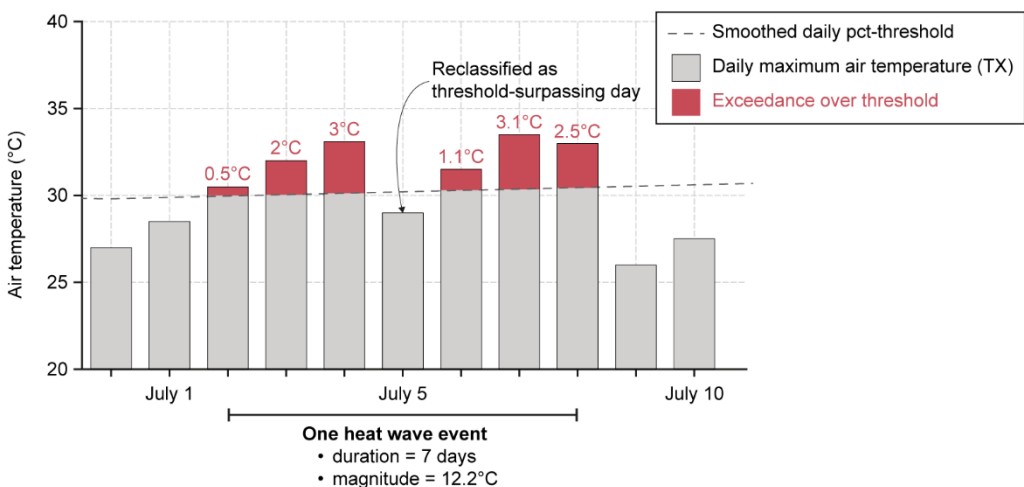

**Figure 4. Percentile-based methodology for detecting heat waves at grid point level.** Red numbers indicate the exceedance of the daily maximum air temperature (tx) above the smoothed percentile-based threshold, which are summed to determine the event magnitude. A minimum duration of three days is required for classification as a heat wave. If a non-qualifying day falls between threshold-surpassing days, it is reclassified as also being threshold-surpassing. The diagram represents a synthetic time series, and the y-axis does not start at zero. For detecting cold waves, we use daily minimum air temperature (tn) instead.



## 2.3 Output aggregation

240 The hazard- and exposure-related attributes of SHEDIS-Temperature were aggregated into output files at two spatial levels: disno level and subdivision level. At the disno level, the spatial extent is defined by the combined boundaries of all identified impacted subdivisions within a given disno (Figure 5a). At the subdivision level, the spatial extent corresponds to the individual polygon of each subdivision (Figure 5b). Both levels provide the same set of attributes, including:

- Metadata attributes
245 - Temperature attributes averaged over the analysis period
- Extreme daily temperature attributes
- Extreme 3-hourly temperature attributes
- Hazard and exposure attributes from the percentile-based event detection analysis

Additionally, results from the percentile-based threshold analysis at the grid point level are stored in separate files for each 250 disno, with one row per percentile-exceeding event (Figure 5c). Where applicable, the date and location of specific attributes are also saved (e.g., the coordinate pair and date of the warmest 3-hourly air temperature recorded at the grid point level). The full set of attributes provided by SHEDIS-Temperature are provided under the section 4 Data records.

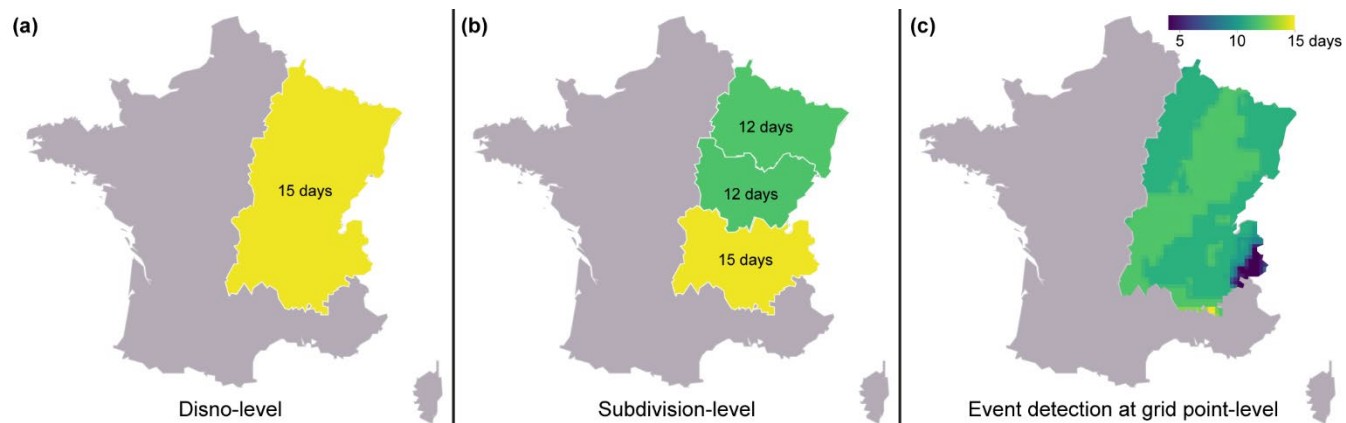

**Figure 5 Illustration of the three output levels of SHEDIS-Temperature, using the maximum duration of a 95th percentile-exceeding**
255 **event in France as an example.** (**a**) At the disno level, the spatial extent (i.e. geometry for analysis) encompasses the combined boundaries of all identified impacted subdivisions, with the maximum duration representing the highest value among the three subdivisions. (**b**) At the subdivision level, the spatial extent corresponds to an individual impacted subdivision, where the maximum duration reflects the longest duration at any grid point within that specific polygon. (**c**) At the grid-point level, events are detected and recorded at the resolution of individual pixels.



## 3 Results

### 3.1 Global analysis of human exposure to temperature extremes

In building SHEDIS-Temperature, we have successfully quantified a wide range of hazard- and exposure-related attributes from 2,835 administrative subdivisions associated with 382 records of temperature-related disasters. Distinct patterns emerge across continents regarding both extreme temperatures and the populations bearing the brunt of these extremes. For instance, North America, Europe and northern Asia stand out for having experienced very cold extremes in highly populated areas (Figure 6a). In terms of human exposure to hot extremes, India, Pakistan and Bangladesh are notably affected by the combination of very high temperatures and large population numbers (Figure 6c).

The data gap in Africa, the Middle East and Southeast Asia is particularly striking and highlights a broader challenge of international disaster databases. Despite these gaps, our results reveal a global pattern in which warmer administrative subdivisions also tend to be more populated (Figure 6). This trend is particularly pronounced in subdivisions that have experienced heat wave disasters (Figure 6d), emphasizing a critical challenge in the face of climate change. These findings also underscore the importance of integrating data across multiple risk dimensions to better identify and understand risk hotspots globally.

Turning now to the results of the event detection analysis using percentile-based thresholds. For a vast majority of the disnos in SHEDIS-Temperature, we could detect percentile exceeding events within the respective subdivisions and analysis periods. All 139 heat wave disnos in the dataset record at least one pct90 event at grid point level within the defined spatiotemporal boundaries, while 133 disnos also experience pct95 events. For over 70% of the heat wave disnos, pct95 events cover more than half of the analysed area. Similarly, among the 243 cold wave disnos, 233 show pct10 events, and 214 also record pct05 events. For nearly 50% of the cold wave disnos, pct05 events cover more than half of the analysed area. Taken together, these results support the reliability of EM-DAT reports and their recorded locations.

In terms of human exposure to these percentile-exceeding events, India once again emerges as particularly affected. All eleven disnos in our dataset with the highest number of person-days exposed to pct95 events are recorded in India. Thereafter follows two heat wave records from the United States (1998 and 2011) and the 2003 heat wave in Germany. Despite not being prone to the lowest absolute temperatures, India also ranks prominently for person-days exposed to pct05 cold waves, accounting for five of the top ten disnos in our sample. Other highly ranked cold wave disnos include events in China (2011), Germany, Bangladesh, France, and Poland (all in 2012).

**Figure 6 Geographical distribution of human exposure to extreme temperatures. (a)** The lowest apparent temperatures recorded within the analysis period for each cold wave disno in the dataset. Each dot represents a subdivision, with color indicating temperature and size
290 representing the total population of the subdivision the year of the record. **(b)** Scatter plot showing the relationship between population and temperature. Each dot represents a subdivision, with color indicating the continent. **(c)** and **(d)** are analogous to **(a)** and **(b)** but depict the highest apparent temperatures in heat wave-impacted subdivisions instead. The x-axes in **(b)** and **(d)** are logarithmic for visualization purposes.

### 3.2 A case study from the fatal European heat waves in 2003

295 As previously noted, the year 2003 stands out as one of the years with the highest number of heat wave disnos in the dataset, driven by widespread and severe European heat waves. Four of the five disnos in our sample with the highest reported fatalities in EM-DAT correspond to this event (all disnos beginning with 2003-0391): Italy (20,089 deaths), France (19,490 deaths),



Spain (15,090 deaths), and Germany (9,355 deaths)[2]. Among these, the French disno is the most severe fatal impact record for which EM-DAT also includes information on its physical magnitude (maximum temperature of 43°C) as well as start and end dates (August 1–20). We now examine how this information aligns with the attributes provided by SHEDIS-Temperature (Figure 7).

The highest 3-hourly air temperature recorded at the grid point level within the impacted subdivisions, based on MSWX data, is 41°C on August 12. This is slightly lower than the magnitude reported by EM-DAT, presumably due to the inherent limitations of gridded datasets, which may miss localized extreme values within individual grid cells. The highest temperatures were recorded in inland France, particularly in Haute-Vienne (Figure 7a-b). However, there is considerable spatial variation within subdivisions, especially in the mountainous areas of the French Alps and the Pyrenees.

Almost all grid points within the analysed subdivisions recorded pct95 events during the analysis period, with a median duration of 11 days (Figure 7c-d). The longest duration, 31 days, was recorded on the island of Corsica, while the shortest duration, lasting only a few days, occurred in coastal Brittany. Across the entire geometry, pct95 events occurred from July 25 to August 31. Regarding human exposure, the total population in these subdivisions was 43 million in 2003, according to GHS-POP data. The total number of person-days exposed to pct95 events amounted to 508 million, with the largest numbers recorded in the region of Île-de-France, which includes Paris (Figure 7e-f).

---

[2] The only disno with a higher reported fatality count is a Russian heatwave disno from 2010, with over 55,000 fatalities recorded.

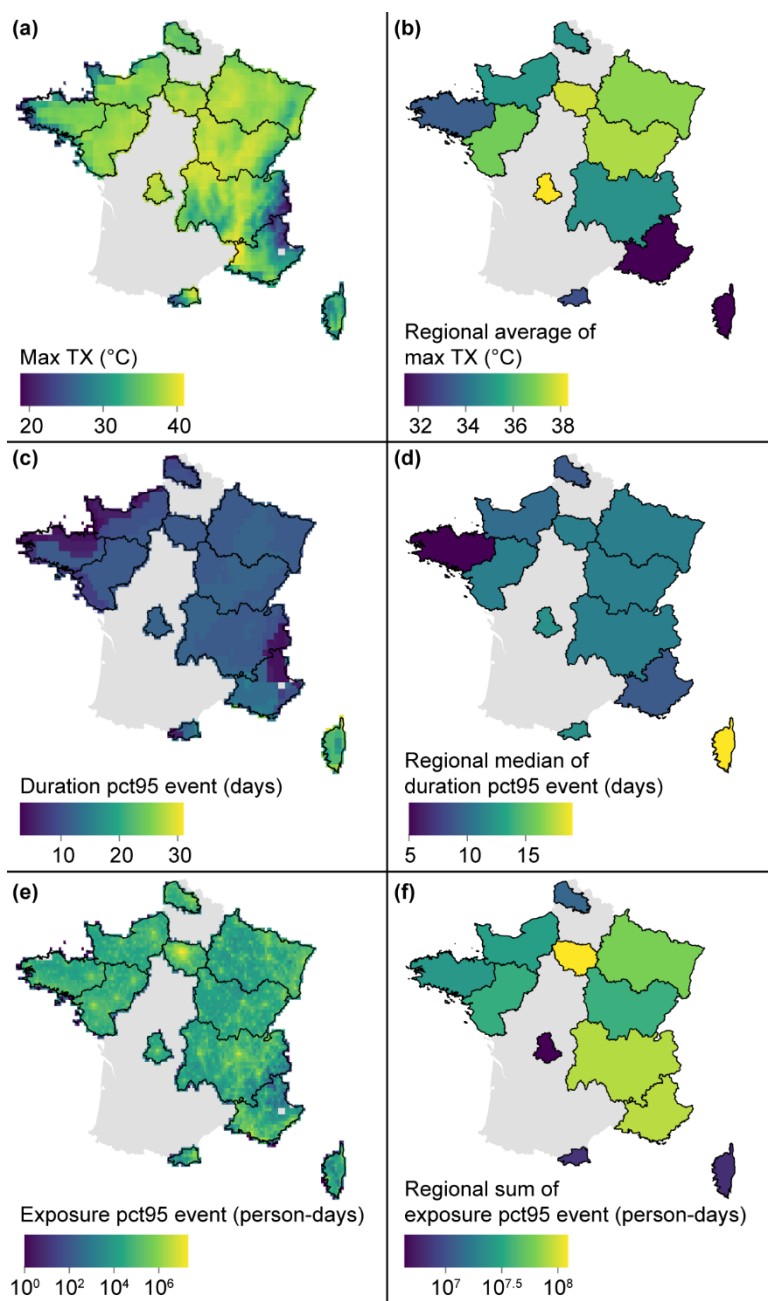

**Figure 7 Selected SHEDIS-Temperature attributes for the 2003 heat wave in France (disno 2003-0391-FRA).** Panels depict the maximum 3-hourly air temperature **(a)-(b)**, duration of pct95 events **(c)-(d)** and human exposure expressed in person-days **(e)-(f)**. The left panels show grid point-level data, while the right panels present outputs at subdivision-level. Note that palette ranges vary across panels, and logaritmic scales are applied for person-days.



### 3.3 Technical validation

#### 3.3.1 Sample coverage

The dataset includes approximately 80% (382 out of 468) of the heat wave and cold wave disnos recorded in EM-DAT between 1979 and 2018, covering 243 of 293 cold waves and 139 of 175 heat waves. This coverage is limited to disnos that have been
geocoded in GDIS, which primarily depends on the level of detail in the original location descriptions in EM-DAT (Rosvold and Buhaug, 2021). Additionally, a small number of subdivisions have been excluded due to geocoding errors in GDIS, and one misplaced GDIS centroid has been manually corrected (see Section 2.1).

Our approach verifies the consistency of GDIS with EM-DAT and eliminates dependence on GDIS-provided ISO codes, which are not completely consistent – with certain countries having been assigned multiple ISO codes in GDIS. While these steps
result in partial coverage of EM-DAT disnos and may lead to occasional omissions of subdivisions, they enhance the overall reliability of our dataset by ensuring a high degree of confidence in the included cases.

#### 3.3.2 Reliability of hazard attributes

We use data from MSWX-Past, a bias-corrected and downscaled version of ERA5. ERA5 shows reduced performance in areas with sparse in situ observations, as it integrates remotely sensed and ground-based measurements (Hersbach et al., 2020; Liu
et al., 2024), and MSWX likely shares this shortcoming. However, since most disaster reports in SHEDIS-Temperature originate from regions with dense station networks (e.g. Europe), this limitation is unlikely to have a substantial impact on SHEDIS-Temperature.

Accuracy is also often reduced in regions with complex terrain and high altitudes. For example, ERA5 tends to underestimate wind speeds in mountainous regions (Beck et al., 2022) which may affect the modelled apparent temperature attributes in
SHEDIS-Temperature. Additionally, errors in 2-m air temperature tend to be larger in regions with complex topography and high elevations, as well as deserts and tropical rainforests (Beck et al., 2022; Liu et al., 2024). This inevitably influences the quality of MSWX. The validation study by Beck et al. (2022) showed that MSWX performs comparably to ERA5 over flat terrain and outperforms ERA5 in high-relief areas. This highlights the advantage of using a high-resolution resampled variant of ERA5 to capture local variations in temperature extremes, as exemplified in Figure 7.

On the topic of data resolution, global gridded datasets are best suited for analyses at global to regional scales. This consideration guided our decision to restrict SHEDIS-Temperature to level 1 and level 2 administrative subdivisions. We reiterate that localized extreme temperature events occurring at spatial scales smaller than the grid resolution may not be fully captured. Additionally, the averaging from 1-hour time steps in ERA5 to the 3-hourly time steps in MSWX may also veil short-lived temperature extremes.





To assess the reliability of the hazard attributes in SHEDIS-Temperature, we systematically compare the most extreme 3-hourly MSWX temperatures recorded at the grid point level with the temperatures reported in EM-DAT. The latter correspond to a maximum temperature for heat waves and a minimum temperature for cold waves. This information is, however, only available for a subset of cases, specifically 94 heat wave disnos and 120 cold wave disnos in our sample (Figure 8).

Overall, the agreement between MSWX and EM-DAT is stronger for heat waves (MAE=2.6°C) than for cold waves
(MAE=8.3°C). For heat waves, MSWX and EM-DAT show reasonable consistency, though EM-DAT values tend to be slightly higher, with an average bias of 0.81°C (Figure 8a). This discrepancy may reflect the inability of gridded datasets like MSWX to fully capture localized temperature extremes, as previously discussed.

A few outliers are evident. For example, a heatwave in the Borno Province, Nigeria, in June 2002 records a maximum temperature of 60°C in EM-DAT, whereas the corresponding MSWX estimate is 44°C (Figure 8b). To put these values into
context, according to the World Meteorological Organisation the official highest registered air temperature on Earth is 56.7°C, recorded in Death Valley in the United States (WMO Records of Weather and Climate Extremes, 2025). This casts doubts on the veridicity of the EM-DAT record, which likely echoes news reporting from the time describing temperatures reaching 55-60°C in Maiduguri (The New Humanitarian, 2002). While the EM-DAT value may be an overestimation of the actual conditions, differences between the two datasets may also reflect the spatial resolution of MSWX, which could miss localized
extreme temperatures, such as those arising due to urban heat island effects.

The comparison for cold waves exhibits greater variability (Figure 8a). On average, EM-DAT reports minimum temperatures 1.8°C higher than MSWX estimates, but the spread is substantial in both directions. For instance, a group of cold wave disnos in India show a stark contrast, with MSWX minimum temperatures near -40°C, while EM-DAT reports values slightly above 0°C (Figure 8a). In December 2002 (disno 2002-0818-IND), for instance, EM-DAT reports a magnitude of 5°C across a
number of regions encompassing the most northern part of India (Bihar, Uttar Pradesh, Himachal Pradesh, Rajasthan, Jharkhand, Jammu and Kashmir, Punjab, Haryana, Delhi provinces). Within these subdivisions, MSWX estimates a minimum of -42°C, recorded in the mountainous Jammu and Kasmir region, which is climatically diverse across altitude levels and regularly experiences sub-zero temperatures in winter.

This discrepancy is likely driven by factors beyond differences in temperature measurement methods. The EM-DAT magnitude
record of 5°C for the 2002 cold wave likely comes from accounts such as "*On many occasions the average temperature was less than 5°C for consecutive days*" (Samra et al., 2003, p. "Preface"). EM-DAT magnitude records may thus, in some cases, reflect prolonged conditions in areas that suffered large socioeconomic losses (e.g. agricultural damage). In contrast, SHEDIS-Temperature quantifies extremes across all grid points within the impacted subdivisions. The percentile-based event detection analysis at the grid-point level can provide users with a more spatially detailed representation of cold waves in regions with
high climatic variability. Taken together, these findings highlight the need for systematic approaches to linking hazard magnitude estimates with disaster impact records.





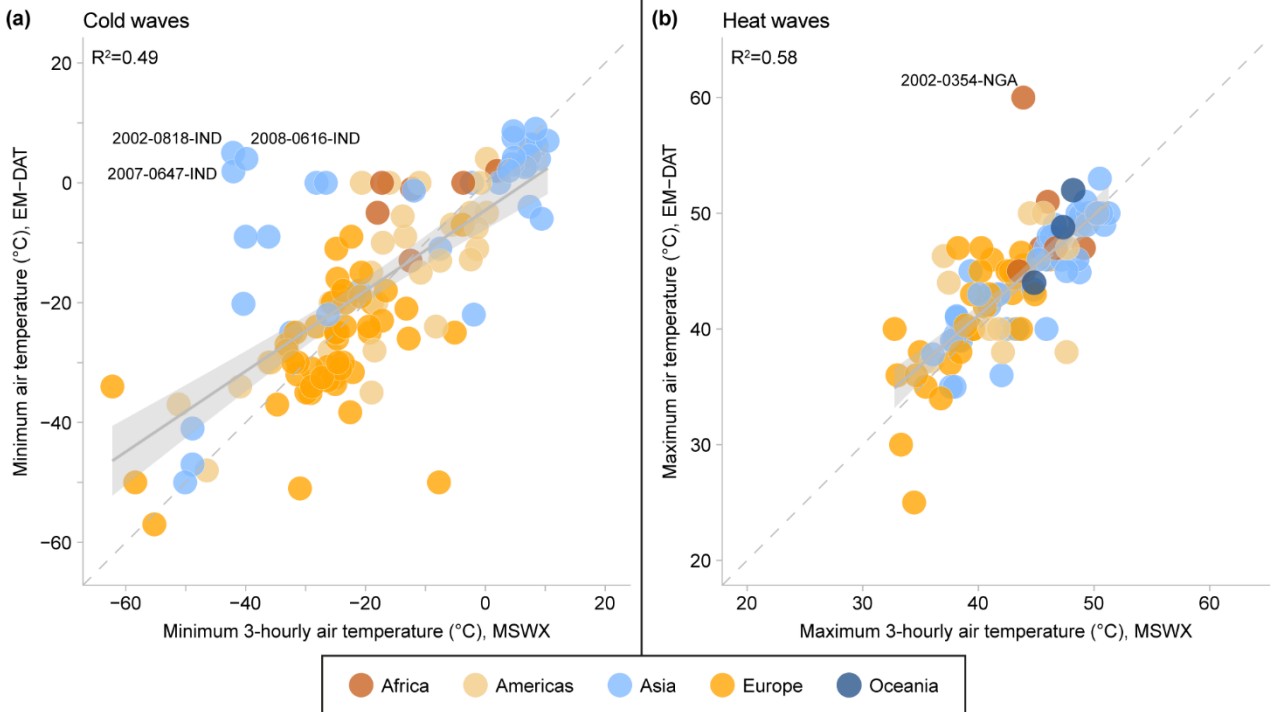

**Figure 8 Comparison of extreme temperatures between EM-DAT and MSWX.** The scatter plots illustrate the relationship between reported temperature extremes in EM-DAT and the corresponding 3-hourly minimum **(a)** and maximum **(b)** temperatures from MSWX for 120 cold waves and 94 heat waves, respectively. Each point represents a disno, with colours indicating the continent.

## 4 Data and code availability

SHEDIS-Temperature is publicly available from a Harvard Dataverse repository (https://doi.org/10.7910/DVN/WNOTTC; Lindersson and Messori, 2025), with replication code published on GitHub (https://github.com/sara-lindersson/shedis-temperature-replication-code; Lindersson, 2025). The dataset is organized into two main folder structures: one for heat waves and one for cold waves (Figure 9). Each folder contains four primary files, with content as outlined in Table 1. Two files (CSV and GeoPackage) contain attributes aggregated at the disno-level, with one row per disno. Two additional files (CSV and GeoPackage) contain attributes aggregated at the subdivision-level, with one row per subdivision and disno. These files do, however, also include information derived at the grid point level. The only distinction between the CSV and GeoPackage files is that the latter also contain the geometries delineating the analysis domain.

SHEDIS-Temperature also includes subfolders containing detailed outputs from the detection of threshold-exceeding events, with subfolder names specifying the threshold and minimum duration used for analysis (Figure 9). These subfolders contain one CSV file per disno for which threshold-exceeding events were detected, with one row per subdivision, coordinate pair, and detected event. The information in these files is detailed in Table 2.



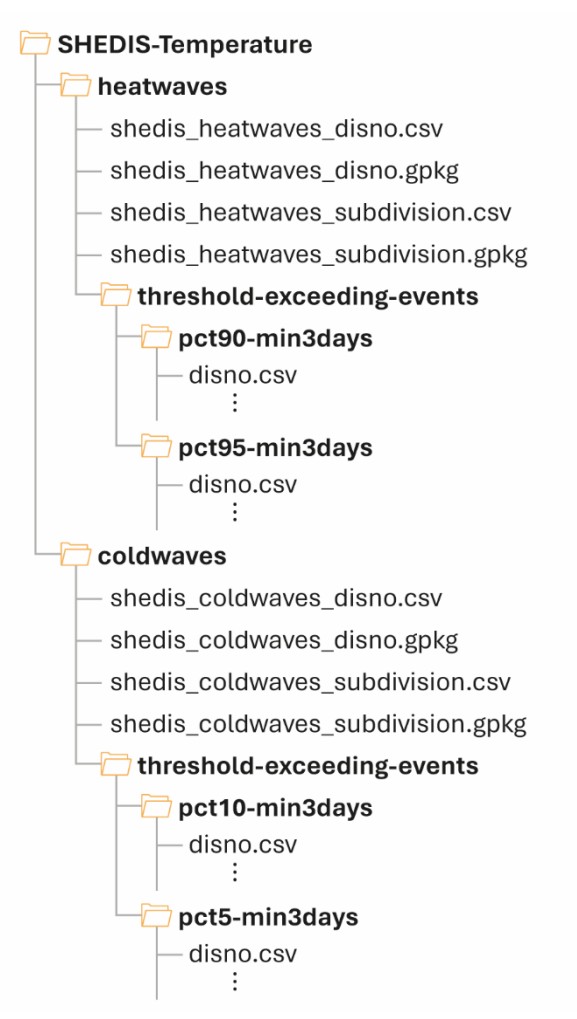


**Figure 9 Folder structure of SHEDIS-Temperature.** The dataset is organized into two main folders, each representing a specific hazard type. Within the the 'threshold-exceeding-events' folders, subfolders are named based on the threshold and minimum duration used for analysis. Files within in these subfolders are named according to their respective disaster identifier number (disno) as provided by EM-DAT (e.g. '2003-0391-FRA').



**Table 1 Attributes available in files beginning with 'shedis'.** These files are provided in both CSV and GeoPackage formats at two spatial levels: the disno level (one row per disno) and the subdivision level (one row per disno and subdivision). This table lists attributes for heat waves in files beginning with 'shedis_heatwaves', including: tx (daily maximum temperature) and atx (daily maximum apparent temperature). For cold waves, these attributes are replaced with tn (daily minimum air temperature) and atn (daily minimum apparent temperature). Attributes from the event detection analysis are here denoted as 'pctXX', with pct90 and pct95 applied to heat waves, and
pct10 and pct05 applied to cold waves. Variables prefixed with 'xy_' represent values calculated at the grid-point level, while all other hazard-related variables have been spatially averaged over the corresponding geometry. An asterisk (*) denotes attributes available exclusively in the GeoPackage format.

| Type | Attribute | Unit | Description |
|---|---|---|---|
| Metadata | iso3c | - | ISO-3 character country code. |
| | country | - | Country name. |
| | disno | - | Disaster identifier number in EM-DAT. |
| | subtype | - | Hazard subtype in EM-DAT (heat wave or cold wave) |
| | gadm_gid | - | Administrative subdivision identifier in GADM. |
| | gadm_level | - | Administrative level of subdivision in GADM (province/equivalent=1, county/district/equivalent=2). |
| | gadm_name | - | Administrative subdivision name in GADM. |
| | geometry* | - | Simplified polygon of administrative subdivision boundaries. |
| | geometry_area_km2 | km$^2$ | Area of "geometry". |
| | geometry_pop | persons | Population total of administrative subdivision the year of "analysis_start". |
| | analysis_start | - | Start date period of analysis. |
| | analysis_end | - | End date period of analysis. |
| Attributes representing temporal averages across the entire period of analysis | mean_t | °C | Average daily-mean air temperature. |
| | mean_at | °C | Average daily-mean apparent temperature. |
| | mean_tx | °C | Average daily-max air temperature. |
| | mean_atx | °C | Average daily-max apparent temperature. |
| The most extreme daily mean values recorded within the period of analysis. | max_t \| max_t_date | °C \| date | Maximum daily-mean air temperature, with recording date. |
| | xy_max_t \| xy_max_t_date \| xy_max_t_coord | °C \| date \| ° | Maximum daily-mean air temperature at grid point-level, with recording date and coordinates. |
| | max_at \| max_at_date | °C \| date | Maximum daily -mean apparent temperature, with recording date. |
| | xy_max_at \| xy_max_at_date \| xy_max_at_coord | °C \| date \| ° | Maximum daily -mean apparent temperature at grid point-level, with recording date and coordinates. |
| The most extreme 3-hourly values recorded within the period of analysis. | max_tx \| max_tx_date | °C \| date | Maximum daily-max air temperature, with recording date. |
| | xy_max_tx \| xy_max_tx_date \| xy_max_tx_coord | °C \| date \| ° | Maximum daily-max air temperature at grid point-level, with recording date and coordinates. |
| | max_atx \| max_atx_date | °C \| date | Maximum daily-max apparent temperature, with recording date. |
| | xy_max_atx \| xy_max_atx_date \| xy_max_atx_coord | °C \| date \| ° | Maximum daily-max apparent temperature at grid point-level, with recording date and coordinates. |
| Hazard and exposure variables from the threshold analysis at grid point level | pctXX_area_percentage | % | Percentage of geometry area that experienced at least one event. |
| | pctXX_pop | persons | Population that experienced at least one event. |
| | pctX_persondays | person-days | Population exposure of events. |
| | pctXX_median_duration | days | Median duration of all events detected within the geometry. |
| | pctXX_max_duration | days | Duration of the longest event detected within the geometry. |
| | pctXX_days | days | Total number of days during which at least one grid point experienced an event. |





| | pctXX_dates | - | List of dates for which at least one grid point in the geometry experienced an event. |
|---|---|---|---|





**Table 2 Attributes in CSV files in the folders named 'threshold-exceeding-events'.** The files are named according to their respective disaster identifier number (disno) from EM-DAT and contain one row per subdivision and event. Each event is identified using percentile-based thresholds at grid point-level. This table lists attributes for heat waves, including tx (daily maximum air temperature) and atx (daily maximum apparent temperature). For cold waves, these attributes are replaced with tn (daily minimum air temperature) and atn (daily minimum apparent temperature). Results are provided for thresholds of analysis using the 90th and 95th percentiles for heat waves and the
10th and 5th percentiles for cold waves, as explicitly indicated in the subfolder names and the 'mean_pctXX'-attribute.

| Type | Attribute | Unit | Description |
|---|---|---|---|
| Metadata | disno | - | Disaster identifier number in EM-DAT. |
| | subtype | - | Hazard subtype in EM-DAT (heat wave or cold wave). |
| | gadm_gid | - | Administrative subdivision identifier in GADM. |
| | gadm_level | - | Administrative level of subdivision in GADM (province/equivalent=1, county/district/equivalent=2). |
| | gadm_name | - | Administrative subdivision name in GADM. |
| | analysis_start | date | Start date period of analysis. |
| | analysis_end | date | End date period of analysis. |
| | x | ° | Longitude of grid point. |
| | y | ° | Latitude of grid point. |
| | cf | - | Coverage fraction (i.e. share of) of grid cell located within subdivision boundaries. |
| | area_km2 | km$^2$ | Area of grid point, scaled with "cf". |
| | pop | persons | Population of grid point the year of the event, scaled with "cf". |
| Overarching event information | event_start | date | Start date of the event |
| | event_end | date | End date of the event |
| | mean_pctXX | °C | Average percentile-based threshold across the event days. |
| | duration | days | Event duration. |
| | persondays | person-days | Human exposure of event at grid point-level by multiplying "pop" with "duration". |
| | magnitude | °C | Temperature exceedance over or below threshold summed across all event days. |
| Temporal averages across all event days. | mean_t | °C | Average daily-mean air temperature. |
| | mean_at | °C | Average daily-mean apparent temperature. |
| | mean_tx | °C | Average daily-max air temperature. |
| | mean_atx | °C | Average daily-max apparent temperature. |
| The most extreme daily mean values recorded within the event period. | max_t \| max_t_date | °C \| date | Maximum daily-mean air temperature, with recording date. |
| | max_at \| max_at_date | °C \| date | Maximum daily-mean apparent temperature, with recording date. |
| The most extreme 3-hourly values recorded within the event period. | max_tx \| max_tx_date | °C \| date | Maximum daily-max air temperature, with recording date. |
| | max_atx \| max_atx_date | °C \| date | Maximum daily-max apparent temperature, with recording date. |



## 4.1 Usage notes

SHEDIS-Temperature extends the international disaster database EM-DAT, which is known to have reporting biases, with

higher coverage in advanced economies such as those in Europe and North America (CRED and UCLouvain, 2024b; Gall et al., 2009; Osuteye et al., 2017). Users should consider this bias when comparing disaster frequencies across continents. Furthermore, EM-DAT records only major disasters that meet at least one of the following criteria: ≥10 fatalities, ≥100 affected individuals, a declared state of emergency, and/or a request for international assistance. The database's coverage is thus affected by exposure and vulnerability, as well as differing national criteria to declare a state of emergency.

The meteorological and population attributes in SHEDIS-Temperature are derived from global gridded products. As such, the results should be interpreted with caution at local scales. Consequently, SHEDIS-Temperature includes administrative subdivisions at the level-1 (province/equivalent) and level-2 (county/district/equivalent) scales, but not finer.

Our dataset is provided at two spatial levels: the disno-level and subdivision-level. We anticipate that the disno-level data will be particularly useful for comparative analyses across countries, while the subdivision-level data will facilitate the examination

of within-country variations.

The spatial boundaries of the SHEDIS-Temperature analysis are limited to the administrative subdivisions recorded as impacted locations in EM-DAT, where 'impacted' refers to areas affected by socioeconomic losses. As a result, these boundaries are not meant to outline the spatial extent of the meteorological events per se. These boundaries also outline the domain for the analyses at grid point level.

For the percentile-based detection analysis, we enforce a minimum duration of three consecutive days for heat waves and cold waves, a widely used criterion in extreme temperature studies. This is more conservative than the EM-DAT's definition, which typically considers events lasting two days or longer (CRED and UCLouvain, 2024b).

We highlight below some key practical usage points to note:

- To link SHEDIS-Temperature with EM-DAT, users can match the disaster identifier code ('disno') present in both
datasets, but in EM-DAT written as 'DisNo.'.
- Users should ensure UTF-8 encoding is used when reading SHEDIS files to correctly display location names.
- For projecting coordinate-specific CSV outputs to raster files, users should adopt the same grid as MSWX.
- The polygons in the GeoPackage files in SHEDIS-Temperature are simplified versions of the original polygons from GADM v3.6. To access the original polygons, users may retrieve the 'gadm_gid' identifiers in SHEDIS-Temperature,
which correspond to 'GID_1' for level-1 subdivisions and 'GID_2' for level-2 subdivisions in GADM.
- The R scripts used to generate SHEDIS-Temperature outputs are available on GitHub as R Markdown files, along with an accompanying ReadMe file.

## 5 Conclusions

International databases of socioeconomic disaster impacts are essential for disaster risk research, yet they display important geographic coverage biases. The data gap is particularly striking in Africa, the Middle East and Southeast Asia, and addressing it will require continued efforts from the global disaster research community. Nonetheless, it is critical to maximize the usefulness of the data that we do have available. SHEDIS-Temperature addresses this need by enriching the information about major temperature-related disasters across five continents.

By providing detailed hazard information – such as temperature thresholds, duration, and geographic distribution – and linking it to exposure data (e.g., population counts during threshold-exceeding events), SHEDIS-Temperature enables more comprehensive analyses of past temperature-related disasters. For instance, users may calculate mortality rates by combining EM-DAT's fatality numbers with the exposure information in SHEDIS-Temperature. Researchers can further combine SHEDIS-Temperature with other socioeconomic and political indicators. This type of information is essential for statistical studies of how risk varies across time and regions. Ultimately, we think that this also can enhance the understanding of social
and societal vulnerabilities, revealing how exposure to extreme temperatures intersects with socioeconomic factors over time and across regions.

At first glance, the results from SHEDIS-Temperature evidence a concerning trend: more populated subdivisions tend to face higher temperatures, a pattern that will likely intensify as climate change progresses. The intersection of rising temperatures and population growth will amplify risk, particularly in regions already facing the most severe temperature-related disasters.
Identifying such risk hotspots underscores the importance of collecting data across the entire disaster risk spectrum in a systematic manner, and of making the outputs accessible to an interdisciplinary set of disaster researchers.

## Author contributions

**SL:** Conceptualization (equal); Data curation (lead); Formal analysis and validation (lead); Methodology (lead); Visualization (lead); Writing – original draft preparation (lead); Writing – review & editing (lead). **GM:** Conceptualization (equal); Funding
Acquisition (lead); Methodology (supporting); Writing – review & editing (supporting).

## Acknowledgements

We gratefully acknowledge the Centre of Natural Hazards and Disaster Science (CNDS) and the Swedish Centre for Impacts of Climate Extremes (climes) for their support in enabling this work. We also extend our thanks to the data providers, including the Centre for Research on the Epidemiology of Disasters (EM-DAT), NASA Socioeconomic Data and Applications Center
(GDIS), the GADM project, GloH2O (MSWX) and the European Commission Joint Research Centre (GHS-POP). AI tools, including ChatGPT (OpenAI) and Copilot (Microsoft) have provided occasional assistance with R code writing and manuscript



revisions, but were not used to generate content. In Figure 1 and Figure 9, we acknowledge using the following vector icons from the Noun Project: 'heat' and 'population' by Adrien Coquet, 'location' by Hermawan, 'health care' by fatimahazzahra, and 'folder' by Landan Lloyd.

### Financial support

- The Centre of Natural Hazards and Disaster Science (CNDS)
- The Swedish Centre for Impacts of Climate Extremes (climes)
- The Swedish Research Council (Vetenskapsrådet), grant number 2022-06599

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
