# Peer review of "SHEDIS-Temperature: Linking temperature-related disaster impacts to subnational data on meteorology and human exposure"

_Earth System Science Data, 2025_

## Referee Comment (RC4)

**Review of "SHEDIS-Temperature: Linking temperature-related disaster impacts to subnational data on meteorology and human exposure" by Sara Lindersson and Gabriele Messori**

**Over Assessment**

The authors present a newly constructed database of temperature-related disaster impacts, which combines meteorological and demographic inputs to provide detailed hazard and exposure data, which is usefully output on different levels (i.e., country, region and also grid point level for heat/cold wave indicators) to satisfy the requirements of a diverse range of end-users.

I was impressed with the rigorous methodology used to provide such a comprehensive dataset that will surely prove invaluable for stakeholders involved in risk mitigation and adaptation (e.g., design of resilience strategies). I think the effort put in to remove inconsistencies in for example EM-DAT will be especially appreciated.

The paper is thorough, the methodology is mostly clear, and the results usefully showcase outputs from the dataset. I think the paper is well suited for publication in ESSD and does not require much in the way of changes for the final version.

I have just a few comments concerning the 'Data and methods' section which I think the authors should clarify and provide some additional detail in the manuscript to ensure the approach is easily followed. These specific (line-by-line) comments are provided below:

**Specific Comments**

**L108:** What does GADM stand for? Should this acronym be spelled out here?

**L118-119:** I find this difficult to conceptualise. Perhaps it could be explained first what the difference between GADM polygons and GDIS subdivisions is? And how the former yields the latter?

Using France as an example, how does the administrative subdivisions (demarcated in red; https://gadm.org/download_country.html) differ to those in Fig. 1C? They look much the same to me.

**L210-211:** If I understood correct, linear detrending is only performed on a copy of the data for percentile estimation. These values are then carried across when detecting heat/cold waves on the non-detrended data.

I wondered what the sensitivity is in detecting extremes if no detrending was instead performed when calculating the daily percentiles? Presumably if there is a statistically significant upward trend, the values for earlier and later years would largely cancel out anyway when calculating a mean over all years.

If the rate of warming is however not uniform over time (e.g., no trend over first 20 years but strong upward trend for the last 10 years), then I could envisage a greater impact on the resultant percentile values (detrended v non-detrended). Perhaps the justification for detrending the data for percentile estimation could be expanded to point this out, assuming this thinking is correct?

**L213-214:** Another sentence I'm not sure I understood. 'Adding back the temporal mean of the daily maximum and minimum time series to the detrended values' → Is this to help preserve the shape of the seasonal cycle which might be lost when detrending? The temporal mean being the daily averages over 30 years (and not the length of the moving window I assume?). Further to my last point that the earlier and later years might often largely cancel out, adding these (non-detrended) values back to the detrended values is therefore presumed not to lead to any inconsistencies?

Some additional clarification I think is warranted here, as I think its very difficult to follow what the authors have done exactly.

---

## Author Comment (AC5)

**Responses to Anonymous Referee #3**

**Referee #3:** This data paper introduces SHEDIS-Temperature, a curated dataset linking EM-DAT national disaster records for heat and cold waves to subnational geometries, meteorological data, and population exposure. The dataset is useful, well-structured, and has clear potential for cross-national hazard–exposure analysis, model benchmarking, and policy applications. Overall, the manuscript is strong, but several clarifications and additional details would improve transparency, reproducibility, and usability.

**Authors**: We sincerely thank Referee #3 for the constructive and encouraging feedback and are pleased to read that the dataset is considered useful for a wide range of applications. We also agree with the provided suggestions and believe they will help further improve the clarity and transparency of the manuscript. Please find our detailed responses to the specific comments below.

**Major comments**

1. **Referee #3:** Abstract: The abstract should include some key evaluation results (e.g., mean absolute error between EM-DAT and MSWX extremes) to convey dataset reliability at first glance.
   **Authors:** Thank you for this suggestion. We agree, and will include key evaluation results in the Abstract in the revised manuscript. However, we are cautious about presenting the comparison statistics between EM-DAT and MSWX as direct indicators of dataset reliability, since the temperatures reported in EM-DAT cannot strictly be considered ground truth.

2. **Referee #3:** Detrending procedure: The manuscript should explicitly clarify what "detrending" means—whether it refers to removing the long-term climatological trend or the seasonal cycle. Equations or a concise methodological description would help. Please also discuss the sensitivity of detrending results to the choice of reference period.
   **Authors**: Thank you for highlighting the need to clarify this. In our study, detrending refers to the removal of the long-term trend from the daily maximum and minimum temperature time series at the grid-point level. The seasonal cycle is preserved. To do this, we applied the *detrend*-function of CDO (Schulzweida, 2023) which removes the long-term trend estimated via least-squares regression. After detrending, the temporal mean of the original series was added back to preserve the baseline level, since the output from the detrending-function is centred around zero. We will include the relevant equations in the revised manuscript. We also note that we need to clarify in the revised version that the detrending was applied to the full time series (1979–2018), whereas the 30-year reference period (1981–2010) was used specifically for percentile calculation.

3. **Referee #3:** Advances over EM-DAT: Although Figure 1 touches on this, the global advances in spatial coverage and finer geometries relative to EM-DAT are not clearly visualized. A global map showing EM-DAT vs. SHEDIS coverage would highlight the added value.
   **Authors**: Thank you, we agree that the advances in spatial detail could be illustrated more clearly. Our initial reason for not including such a visualization was that the GDIS article (Rosvold and Buhaug, 2021) already presents this in an effective way. However, we agree that it would still be valuable to provide a visualization tailored to our subsample of EM-DAT records included in SHEDIS. We will therefore revise Figure 1 accordingly in the updated manuscript.

4. **Referee #3**: Choice of MSWX: The manuscript should justify why MSWX was selected as the meteorological input, rather than ERA5-Land, which has the same spatial resolution. A short rationale (e.g., bias corrections, variable availability) is needed.
   **Authors**: Thank you for pointing this out, which was also noted by Referee #1. We agree and will add a concise explanation in the revised manuscript to clarify our choice of MSWX over ERA5-Land.

5. **Referee #3:** Cross-comparison with independent datasets: The study compares EM-DAT records with MSWX-derived extremes, but this is not fully independent from SHEDIS. A cross-check with another dataset (e.g., E-OBS, GHCN, Berkeley Earth, or reanalyses) would provide an independent validation.
   **Authors:** Thank you for highlighting this important point. We agree that cross-checking with an independent dataset further strengthens the quality assessment. The article introducing MSWX (Beck et al., 2022) provides a global validation against station observations, and we will refer to those results in the manuscript (together for the rationale for choosing MSWX). In addition, we will complement this by conducting our own comparison with E-OBS daily maximum (for heat waves) and minimum (for cold waves) temperatures for the European records in our sample, to quantify the ability of MSWX to capture extremes.

6. **Referee #3:** Apparent temperature: Provide more detail on how apparent temperature was calculated (equations, inputs). This is important for reproducibility and comparability with alternative indices such as UTCI or WBGT.
   **Authors:** We agree. We used the apparentTemp-function by the R package HeatStress (https://github.com/anacv/HeatStress). We will provide more details on this, including inputs and equations, in the revised manuscript.

7. **Referee #3:** Area vs. geo-projection: Tables define variables in km², but the gridded input is in WGS84. Clarify whether the data were reprojected or area-corrected to ensure comparable cell areas across latitudes.
   **Authors:** We agree that this needs to be reported as well, will clarify it in the revised manuscript. For calculating area of grid cells, we used the cellsize-function by the R package terra (https://rspatial.github.io/terra/) and for calculating polygons we used the st_area-function by the R package sf (https://r-spatial.github.io/sf/index.html). Both of these function perform area-corrected calculations if the input is in a geographic CRS like WGS84.

8. **Referee #3:** Percentile thresholds: Provide references for the use of a 31-day window for percentile determination.
   **Authors:** Thanks for highlighting this. We will provide references, including Russo et al. (2015) and Vogel et al. (2019).

9. **Referee #3:** Minimum duration: Provide references or justification for the choice of a three-day minimum duration for events.
   **Authors:** Thank you for noting this. We will clarify the rationale and references behind this choice in the revised manuscript. We chose this minimum duration since it is widely used in the climate literature (e.g. Meehl and Tebaldi, 2004; Perkins and Alexander, 2013; Perkins-Kirkpatrick and Lewis, 2020). While these kinds of thresholds will always be, to some extent, arbitrary we think that the main benefit here is the application of consistent methodological choices across all records to ensure comparability. We will also further highlight that users who prefer different event detection settings can use our publicly available R-scripts to do so.

10. **Referee #3:** Uncertainty guidance: There is no quantified uncertainty guidance, required by ESSD.

    **Authors:** Thank you for highlighting this limitation. We will include a clear and condense section on uncertainty guidance to support users. This will cover limitations of the parent database EM-DAT (such as inclusion criteria and known biases), key findings from our validation and comparison analyses (with E-OBS and EM-DAT's temperature data), and the potential omission of local effects (e.g., urban heat islands). In doing so, we will emphasize that the dataset is best suited for analyses at regional to international scales, while more detailed data may be preferable for local applications.

**Minor comments**

**Referee #3:**

- Correct minor typos:
    - "logaritmic" → "logarithmic"
    - "recrods" → "records"
    - "percieved" → "perceived"
    - "Jammu and Kasmir" → "Jammu and Kashmir"
    - "the the" duplication
    - "Files within in these subfolders" → "Files within these subfolders"
- Spelling: Ensure consistent spelling of "GADM" (some occurrences appear inconsistent).
- Figure 9: Clean up the duplicated words and phrasing in the caption/description.

**Authors:** We thank Referee #3 for also capturing these details, we will amend them in the revised version of the manuscript.

**References provided by Authors:**

Beck, H. E., van Dijk, A. I. J. M., Larraondo, P. R., McVicar, T. R., Pan, M., Dutra, E., and Miralles, D. G.: MSWX: Global 3-Hourly 0.1° Bias-Corrected Meteorological Data Including Near-Real-Time Updates and Forecast Ensembles, Bull. Am. Meteorol. Soc., 103, E710–E732, https://doi.org/10.1175/BAMS-D-21-0145.1, 2022.

Meehl, G. A. and Tebaldi, C.: More Intense, More Frequent, and Longer Lasting Heat Waves in the 21st Century, Science, 305, 994–997, https://doi.org/10.1126/science.1098704, 2004.

Perkins, S. E. and Alexander, L. V.: On the Measurement of Heat Waves, J. Clim., 26, 4500–4517, https://doi.org/10.1175/JCLI-D-12-00383.1, 2013.

Perkins-Kirkpatrick, S. E. and Lewis, S. C.: Increasing trends in regional heatwaves, Nat. Commun., 11, 3357, https://doi.org/10.1038/s41467-020-16970-7, 2020.

Rosvold, E. L. and Buhaug, H.: GDIS, a global dataset of geocoded disaster locations, Sci. Data, 8, 61, https://doi.org/10.1038/s41597-021-00846-6, 2021.

Russo, S., Sillmann, J., and Fischer, E. M.: Top ten European heatwaves since 1950 and their occurrence in the coming decades, Environ. Res. Lett., 10, 124003, https://doi.org/10.1088/1748-9326/10/12/124003, 2015.

Schulzweida, U.: CDO User Guide, 2023.

Vogel, M. M., Zscheischler, J., Wartenburger, R., Dee, D., and Seneviratne, S. I.: Concurrent 2018 Hot Extremes Across Northern Hemisphere Due to Human-Induced Climate Change, Earths Future, 7, 692–703, https://doi.org/10.1029/2019EF001189, 2019.

---

## Author Response (AR1)

**Responses to Anonymous Referee #1**

**Referee #1:** The authors have presented a useful dataset with clear explanations of the processing. The dataset allows for the fair comparison of extreme heat and cold events across the globe, subject to the geographical biases inherent in the available data. The dataset clearly provides added value to the EM-DAT database.

I have a suggestion and also a question.

**Authors:** The author team thank referee #1 for providing a positive and constructive comment, we are happy to read that our dataset is found to be useful. Please find answers to the individual points of improvement below.

- Referee #1: Firstly, in section 2.2.4 the authors could state the use of a moving average window earlier in the first paragraph to avoid confusion
   Authors: We agree with and have revised the first paragraph of section 2.2.4 accordingly.
- Referee #1: Secondly, I'd like to know why MSWX was chosen over ERA5-LAND? It isn't clear to me from the text what advantage MSWX offers over the hourly 0.1 degree scale reanalysis provided by ERA5-LAND.
  Authors: We agree that ERA5-Land is also a viable data source for this type of work. Our decision to use MSWX was based on both our own analysis and due to practical reasons. First, we compared maximum and minimum temperature estimates from MSWX and ERA5-Land against EM-DAT records for a subset of our study area. Both products showed broadly similar agreement with EM-DAT, with the main difference being that ERA5-Land aligned less well with minimum temperatures during cold waves. Second, in practical terms, we also found the 3-hourly structure of MSWX less computationally demanding than the hourly files from ERA5-Land. Given these considerations, we chose to proceed with MSWX, while acknowledging that ERA5-Land would also have been a defensible choice. To increase transparency, we would be happy to include the results of our MSWX-ERA5Land-EM-DAT comparison in the revised manuscript.
  - Referee #1: Hi Sara, I don't think it is necessary to include the results of the ERA5LAND comparison, but given it is generally the standard dataset used in these kind of situations, I think it would be good to briefly explain your choice of MSWX as you have done in your response. It is also valuable from the point of view of identifying areas of relative weakness with ERA5-LAND. I would then happily accept the paper for publication.
    Authors: Thank you, Referee #1, for this helpful suggestion. We have modified the text accordingly in the revised manuscript, please see the second paragraph in section 2.2.2.

We are grateful for your supportive feedback and pleased to read your positive assessment of our work.

**Responses to Anonymous Referee #2**

**Referee #2:**

The authors present a newly constructed database of temperature-related disaster impacts, which combines meteorological and demographic inputs to provide detailed hazard and exposure data, which is usefully output on different levels (i.e., country, region and also grid point level for heat/cold wave indicators) to satisfy the requirements of a diverse range of end-users.

I was impressed with the rigorous methodology used to provide such a comprehensive dataset that will surely prove invaluable for stakeholders involved in risk mitigation and adaptation (e.g., design of resilience strategies). I think the effort put in to remove inconsistencies in for example EM-DAT will be especially appreciated.

The paper is thorough, the methodology is mostly clear, and the results usefully showcase outputs from the dataset. I think the paper is well suited for publication in ESSD and does not require much in the way of changes for the final version.

I have just a few comments concerning the 'Data and methods' section which I think the authors should clarify and provide some additional detail in the manuscript to ensure the approach is easily followed. These specific (line-by-line) comments are provided below:

**Authors:** We sincerely thank Referee #2 for the very positive and constructive feedback. We are delighted that our dataset is perceived as useful for a wide range of users. We also appreciate the specific comments, which will help us improve the clarity and transparency of our "Data and Methods" section. Our detailed responses are outlined below.

**Specific Comments**

- Referee #2: L108: What does GADM stand for? Should this acronym be spelled out here?
  - **Authors:** Thank you for noting this. GADM is the official name of the database and stands for *the Database of Global Administrative Areas*. We have restructured a sentence in the third paragraph of section 2.1 to make this clearer.
- Referee #2: L118-119: I find this difficult to conceptualise. Perhaps it could be explained first what the difference between GADM polygons and GDIS subdivisions is? And how the former yields the latter?
  - **Authors:** We agree that this needs to be explained in a more clear way. The polygons of the administrative subdivisions are obtained from GADM. GDIS is a database that links EM-DAT records (impact information) to the corresponding GADM polygons (administrative regions/units) to create a database of administrative regions that have been impacted by the disasters. We have revised paragraphs 3-6 in section 2.1 to explain this in a more accessible way.
- Referee #2: Using France as an example, how does the administrative subdivisions (demarcated in red; https://gadm.org/download\_country.html) differ to those in Fig. 1C? They look much the same to me.
  - **Authors:** You are correct, they are and should be the same. As noted in the comment above, we retain the official boundaries of the administrative units, but quantify hazard and exposure information only for those units identified as impacted by EM-DAT and GDIS. We have specifically added a clarifying sentence on this at the

**end of paragraph 6 in section 2.1**

• Referee #2: L210-211: If I understood correct, linear detrending is only performed on a copy of the data for percentile estimation. These values are then carried across when detecting heat/cold waves on the non-detrended data.

I wondered what the sensitivity is in detecting extremes if no detrending was instead performed when calculating the daily percentiles? Presumably if there is a statistically significant upward trend, the values for earlier and later years would largely cancel out anyway when calculating a mean over all years.

If the rate of warming is however not uniform over time (e.g., no trend over first 20 years but strong upward trend for the last 10 years), then I could envisage a greater impact on the resultant percentile values (detrended v non-detrended). Perhaps the justification for detrending the data for percentile estimation could be expanded to point this out, assuming this thinking is correct?

**Authors:** You have understood our methodology correctly. We agree you raise an important point here, and in the revised manuscript we have tested the influence of the detrending procedure on our outputs, please see the new Appendix C. We have also explained our detrending procedure in a more clear way, please see the updated section 2.2.4.

• Referee #2: L213-214: Another sentence I'm not sure I understood. 'Adding back the temporal mean of the daily maximum and minimum time series to the detrended values' → Is this to help preserve the shape of the seasonal cycle which might be lost when detrending? The temporal mean being the daily averages over 30 years (and not the length of the moving window I assume?). Further to my last point that the earlier and later years might often largely cancel out, adding these (non-detrended) values back to the detrended values is therefore presumed not to lead to any inconsistencies?

Some additional clarification I think is warranted here, as I think its very difficult to follow what the authors have done exactly.

**Authors:** Thank you for highlighting this. When detrending, we use the *detrend*-function in CDO (Climate Data Operators), which removes the linear trend from the time series. After detrending, the series has a mean near zero (since they are the remaining residuals), but we want anomalies to remain relative to the original mean. Therefore, we add back the original temporal mean (calculated over the full time series). This restores the series so that it fluctuates around its original mean while removing the linear trend. We have made this more explicit in the updated version of section 2.2.4, by revising the text and including equations.

**Responses to Anonymous Referee #3**

**Referee #3:** This data paper introduces SHEDIS-Temperature, a curated dataset linking EM-DAT national disaster records for heat and cold waves to subnational geometries, meteorological data, and population exposure. The dataset is useful, well-structured, and has clear potential for cross-national hazard–exposure analysis, model benchmarking, and policy applications. Overall, the manuscript is strong, but several clarifications and additional details would improve transparency, reproducibility, and usability.

**Authors**: We sincerely thank Referee #3 for the constructive and encouraging feedback and are pleased to read that the dataset is considered useful for a wide range of applications. We also agree with the provided suggestions and believe they will help further improve the clarity and transparency of the manuscript. Please find our detailed responses to the specific comments below.

**Major comments**

- Referee #3: Abstract: The abstract should include some key evaluation results (e.g., mean absolute error between EM-DAT and MSWX extremes) to convey dataset reliability at first glance.
  - **Authors:** Thank you for this suggestion. We agree, and have included key evaluation results from the new technical validation assessment, in which we evaluated our MSWX-derived outputs with E-OBS data. Please see lines 22-24 on page 1.
- 2. **Referee #3:** Detrending procedure: The manuscript should explicitly clarify what "detrending" means—whether it refers to removing the long-term climatological trend or the seasonal cycle. Equations or a concise methodological description would help. Please also discuss the sensitivity of detrending results to the choice of reference period.
  - **Authors**: Thank you for highlighting the need to clarify this. In our study, detrending refers to the removal of the long-term trend from the daily maximum and minimum temperature time series at the grid-point level. The seasonal cycle is preserved. To do this, we applied the *detrend*-function of CDO (Schulzweida, 2023) which removes the long-term trend estimated via least-squares regression. After detrending, the temporal mean of the original series was added back to preserve the baseline level, since the output from the detrending-function is centred around zero. We have made this more explicit in the updated section 2.2.4. We also analyse the influence of detrending on our outputs in the new Appendix C.
- 3. **Referee #3:** Advances over EM-DAT: Although Figure 1 touches on this, the global advances in spatial coverage and finer geometries relative to EM-DAT are not clearly visualized. A global map showing EM-DAT vs. SHEDIS coverage would highlight the added value.
  - **Authors**: Thank you, we agree that this is a nice idea to also illustrate the advancement in spatial detail. Our initial reason for not including such a visualization was that the GDIS article (Rosvold and Buhaug, 2021) already presents this in an effective way. We explored revising Figure 1 to include the subnational units, but this made the figure overly cluttered, particularly because the sample contains both level-1 and level-2 administrative units. To address this, we have instead added a new figure (Figure A1 in Appendix A) to illustrate the spatial detail more clearly in the revised manuscript. This is referred to in the end of paragraph 1 of section 2.1.1. When working with this, we also realized that the old version of Figure 1 needed a

- small revision: the legend of panel (a) was not clear in showing how the intervals were divided. We have therefore also revised Figure 1 with regards to this.
- 4. **Referee #3**: Choice of MSWX: The manuscript should justify why MSWX was selected as the meteorological input, rather than ERA5-Land, which has the same spatial resolution. A short rationale (e.g., bias corrections, variable availability) is needed.
  - **Authors**: Thank you for pointing this out, which was also noted by Referee #1. We agree and have added a concise explanation in the revised manuscript to clarify our choice of MSWX over ERA5-Land, please see the second paragraph in section 2.2.2.
- 5. Referee #3: Cross-comparison with independent datasets: The study compares EM-DAT records with MSWX-derived extremes, but this is not fully independent from SHEDIS. A cross-check with another dataset (e.g., E-OBS, GHCN, Berkeley Earth, or reanalyses) would provide an independent validation.
  Authors: Thank you for highlighting this important point. We agree that cross-checking with an independent dataset further strengthens the quality assessment. The article introducing MSWX (Beck et al., 2022) provides a global validation against station observations, and we will refer to those results in the manuscript (together for the rationale for choosing MSWX). In addition, we have complemented this by conducting our own comparison with E-OBS daily maximum (for heat waves) and minimum (for cold waves) temperatures for the European records in our sample, to quantify the ability of MSWX to capture extremes. Please see section 3.3.2.
- 6. Referee #3: Apparent temperature: Provide more detail on how apparent temperature was calculated (equations, inputs). This is important for reproducibility and comparability with alternative indices such as UTCI or WBGT.
  Authors: We agree. We used the apparentTemp-function by the R package HeatStress (<a href="https://github.com/anacv/HeatStress">https://github.com/anacv/HeatStress</a>). We have provided more details on this, including inputs and equations, in the revised manuscript in paragraph 3, section 2.2.2, including Equations (1) and (2).
- 7. Referee #3: Area vs. geo-projection: Tables define variables in km², but the gridded input is in WGS84. Clarify whether the data were reprojected or area-corrected to ensure comparable cell areas across latitudes.
  Authors: We agree that this needs to be reported as well. For calculating area of grid cells, we used the cellsize-function by the R package terra (<a href="https://rspatial.github.io/terra/">https://rspatial.github.io/terra/</a>) and for calculating polygons we used the st\_area-function by the R package sf (<a href="https://r-spatial.github.io/sf/index.html">https://r-spatial.github.io/sf/index.html</a>). Both of these function perform area-corrected calculations if the input is in a geographic CRS like WGS84. We have clarified this in the first paragraph of section 2.
- Referee #3: Percentile thresholds: Provide references for the use of a 31-day window for percentile determination.
   Authors: Thanks for highlighting this. We will have provided two additional references, Russo et al. (2015) and Vogel et al. (2019) regarding this, in the first paragraph of section 2.2.4.
- Referee #3: Minimum duration: Provide references or justification for the choice of a three-day minimum duration for events.
   Authors: Thank you for noting this. We chose this minimum duration since it is widely used in the climate literature (e.g. Meehl and Tebaldi, 2004; Perkins and Alexander, 2013; Perkins-Kirkpatrick and Lewis, 2020). We have added these references in

**paragraph 2, section 2.2.5**

While these kinds of thresholds will always be, to some extent, arbitrary we think that the main benefit here is the application of consistent methodological choices across all records to ensure comparability. And users who prefer different event detection settings can use our publicly available R-scripts to do so. We have further highlighted this in paragraph 6, section 4.1. This was also stated in the last sentence in paragraph 4, section 1.

10. **Referee #3:** Uncertainty guidance: There is no quantified uncertainty guidance, required by ESSD.

**Authors:** Thank you for highlighting this limitation. We will think our newly added technical validation against E-OBS provides improved uncertainty estimates, and we have revised the usage notes (section 4.1) with this respect. The usage guid cover limitations of the parent database EM-DAT (such as inclusion criteria and known biases), key findings from our validation and comparison analyses (with E-OBS temperature data), and the potential omission of local effects (e.g., urban heat islands). In doing so, we have also emphasized that the dataset is best suited for analyses at regional to international scales.

**Minor comments**

**Referee #3:**

- Correct minor typos:
  - o "logaritmic" → "logarithmic"
  - $\circ$  "recrods"  $\rightarrow$  "records"
  - o "percieved" → "perceived"
  - $\circ$  "Jammu and Kasmir"  $\rightarrow$  "Jammu and Kashmir"
  - o "the the" duplication
  - $\circ$  "Files within in these subfolders"  $\rightarrow$  "Files within these subfolders"
- Spelling: Ensure consistent spelling of "GADM" (some occurrences appear inconsistent).
- Figure 9: Clean up the duplicated words and phrasing in the caption/description.

**Authors:** We thank Referee #3 for also capturing these details. They have been amended in the revised version of the manuscript.

**References provided by Authors:**

Beck, H. E., van Dijk, A. I. J. M., Larraondo, P. R., McVicar, T. R., Pan, M., Dutra, E., and Miralles, D. G.: MSWX: Global 3-Hourly 0.1° Bias-Corrected Meteorological Data Including Near-Real-Time Updates and Forecast Ensembles, Bull. Am. Meteorol. Soc., 103, E710–E732, https://doi.org/10.1175/BAMS-D-21-0145.1, 2022.

Meehl, G. A. and Tebaldi, C.: More Intense, More Frequent, and Longer Lasting Heat Waves in the 21st Century, Science, 305, 994–997, https://doi.org/10.1126/science.1098704, 2004.

Perkins, S. E. and Alexander, L. V.: On the Measurement of Heat Waves, J. Clim., 26, 4500–4517, https://doi.org/10.1175/JCLI-D-12-00383.1, 2013.

Perkins-Kirkpatrick, S. E. and Lewis, S. C.: Increasing trends in regional heatwaves, Nat. Commun., 11, 3357, https://doi.org/10.1038/s41467-020-16970-7, 2020.

Rosvold, E. L. and Buhaug, H.: GDIS, a global dataset of geocoded disaster locations, Sci. Data, 8, 61, https://doi.org/10.1038/s41597-021-00846-6, 2021.

Russo, S., Sillmann, J., and Fischer, E. M.: Top ten European heatwaves since 1950 and their occurrence in the coming decades, Environ. Res. Lett., 10, 124003, https://doi.org/10.1088/1748-9326/10/12/124003, 2015.

Schulzweida, U.: CDO User Guide, 2023.

Vogel, M. M., Zscheischler, J., Wartenburger, R., Dee, D., and Seneviratne, S. I.: Concurrent 2018 Hot Extremes Across Northern Hemisphere Due to Human-Induced Climate Change, Earths Future, 7, 692–703, https://doi.org/10.1029/2019EF001189, 2019.